# Dietary Fucoxanthin Induces Anoikis in Colorectal Adenocarcinoma by Suppressing Integrin Signaling in a Murine Colorectal Cancer Model

**DOI:** 10.3390/jcm9010090

**Published:** 2019-12-29

**Authors:** Masaru Terasaki, Mimori Ikuta, Hiroyuki Kojima, Takuji Tanaka, Hayato Maeda, Kazuo Miyashita, Michihiro Mutoh

**Affiliations:** 1School of Pharmaceutical Sciences, Health Sciences University of Hokkaido, 1757 Kanazawa, Ishikari-Tobetsu, Hokkaido 061-0293, Japan; mimoririri.1995@gmail.com (M.I.); hirokojima@hoku-iryo-u.ac.jp (H.K.); 2Cancer Prevention Laboratories, Health Sciences University of Hokkaido, 1757 Kanazawa, Ishikari-Tobetsu, Hokkaido 061-0293, Japan; 3Department of Diagnostic Pathology and Research Center of Diagnostic Pathology, Gifu Municipal Hospital, 7-1 Kashima-cho, Gifu 500-8513, Japan; tmntt08@gmail.com; 4Faculty of Agriculture and Life Science, Hirosaki University, 3 Bunkyo-cho, Hirosaki, Aomori 036-8561, Japan; hayatosp@hirosaki-u.ac.jp; 5Laboratory of Biofunctional Material Chemistry, Division of Marine Bioscience, Graduate School of Fisheries Sciences, Hokkaido University, Hakodate, Hokkaido 041-8611, Japan; kmiya@fish.hokudai.ac.jp; 6Epidemiology and Prevention Group, Center for Public Health Sciences, National Cancer Center, 5-1-1 Tsukiji, Chuo-ku, Tokyo 104-0045, Japan; mimutoh@ncc.go.jp

**Keywords:** anoikis, cancer chemoprevention, carotenoid, colorectal cancer, fucoxanthin

## Abstract

Fucoxanthin (Fx), abundantly contained in edible brown algae, is a carotenoid with strong anti-cancer potential. Anoikis is an anchor-dependent apoptosis particularly related to integrin signaling, and a target for cancer preventive strategies. We recently demonstrated that Fx prevented colon cancer in azoxymethane-dextrane sodium sulfate (AOM/DSS) carcinogenic model mice, and that it increased anoikis-like integrin β1^low/-^/cleaved caspase-3^high^ cells in colonic mucosal crypts. However, an induction mechanism of anoikis by Fx in adenocarcinoma tissue remains unresolved. Thus, we investigated anoikis in colonic adenocarcinoma in AOM/DSS mice. Fx administration (30 mg/kg body weight) significantly suppressed the incidence and multiplicity of colonic adenocarcinoma in AOM/DSS mice. A number of anoikis-like integrin β1^low/-^/cleaved caspase-3^high^ cells in colonic adenocarcinoma and mucosal crypts were significantly increased, 8.3- and 3.5-fold in the Fx group compared with those of the control group, respectively. The results indicated the increase of anoikis-like cells occurred more strongly in colonic adenocarcinoma than in colonic mucosal crypts. In addition, integrin β1 expression, and pFAK (Tyr^397^) and pPaxillin (Tyr^31^) activation in mucosal tissue decreased 0.7-, 0.5- and 0.6-fold by Fx administration, respectively. The results suggest that Fx induces anoikis in colonic adenocarcinoma developed by AOM/DSS treatment through attenuation of integrin signaling.

## 1. Introduction

Anoikis is cell-detachment apoptosis induced by loss of integrin-mediated anchorage from the extracellular matrix (ECM), and it is indispensable for normal embryogenesis and homeostasis. In the anoikis process, PI3K/Akt-, MAPK- and TGF-β signaling are first suppressed by activation of some transmembrane receptors, the cells then detach from the ECM, and finally caspases triggering anoikis are activated. This stepwise apoptosis is inhibited universally in cancer cells. Therefore, cancer cells are able to survive, leading to induction of epithelial-mesenchymal transition (EMT), invasion and metastasis [1,2,3]. Particularly, the cell-matrix receptor of integrin subunits and the activated protein of focal adhesion kinase (FAK) are both overexpressed in many types of cancer and contribute to malignancy [4,5,6,7]. Aberrant regulation of integrin signaling in a tumor microenvironment is associated with enhancement of stemness, EMT and anoikis resistance in cancer epithelial cells, and this situation allows accumulation of inflammatory cells and cancer-associated fibroblasts (CAFs) in tumor tissue [1,8]. Some clinical trials have demonstrated that targeting integrin and FAK is useful in an anti-tumor therapeutic strategy [9,10,11]. However, many clinical trials have failed to show its usefulness [12,13]. 

There has been little evidence available regarding a cancer prevention strategy focusing on anoikis. Some reports demonstrated that anoikis could be induced by food components in vitro. Eicosapentaenoic acid, a dietary polyunsaturated fatty acid abundant in fish and algae, has been suggested to enhance anoikis with G1-arrest in human colorectal cancer HT-29 cells [14]. α-Lipoic acid, contained in green/yellow vegetables, enhances the potency of anti-cancer drugs against H460 lung cancer cells, and induces anoikis with integrin β1/β3 down-regulation [15]. Apigenin, a dietary plant flavonoid, induces anoikis in human melanoma A375 and A2058 cells through integrin and FAK inhibition [16]. Thus, it may be applicable for some food components to be used as cancer chemoprevention agents to induce anoikis. Generally, before conducting an intervention trial, it is necessary to confirm the action of agents in an animal study to establish proof of concept. However, animal studies targeting anoikis induced by dietary compounds have scarcely been concluded. 

Fucoxanthin (Fx), a non-provitamin A carotenoid, is contained in brown marine algae such as *Undaria pinnatifida* (wakame) and *Sargassum horneri* (akamoku). Both brown marine algae are consumed frequently in Japan. Fx is a polyene compound with an allene and a monoepoxide (Figure 1A). Fx has been demonstrated to show polyfunctional features, such as anti-cancer [17,18], anti-inflammation [19], anti-obesity [20] and anti-diabetes [21] in humans and rodents. No serious adverse events have been reported with Fx administration in animal experiments [22,23]. So far, an intervention study of Fx against cancer has not been conducted. Several reports demonstrated that Fx induces apoptosis in many organ types of cancer cells [24,25,26,27,28,29]. The reported cell death process in these cancer cells was largely dependent on Akt, Bcl-2, MAPK, NFκB, and STAT and caspase-3. Interestingly, these molecules also play an important role in anoikis.

Recently, we reported that fucoxanthinol (FxOH), a prime metabolite of Fx, could induce anoikis in human colorectal cancer DLD-1 cells through suppression of integrin signaling [30]. This study suggested that low or negative (low/-) levels of integrin β1 expression, or phosphorylation (p) of FAK(Tyr^397^) or Paxillin(Tyr^31^) with activated (cleaved) caspase-3 (high) could be considered as hallmarks of anoikis during the anoikis process in DLD-1 cells by FxOH. Moreover, Fx administration induced cancer chemoprevention in azoxymethane-dextrane sodium sulfate (AOM/DSS) carcinogenic model mice with significant increases of integrin β1^low/-^/cleaved caspase-3^high^, pFAK(Tyr^397^)^low/-^/cleaved caspase-3^high^ and pPaxillin(Tyr^31^)^low/-^/cleaved caspase-3^high^ cells in colonic mucosal crypts [31]. However, the experimental period in the mice is a brief of 8 weeks after the final DSS exposure and is the timing that few colon adenocarcinoma could be identified. Therefore, it had not been clarified whether Fx induces anoikis in adenocarcinoma tissue.

In the present study, we examined the effect of Fx-induced anoikis on colonic adenocarcinoma in AOM/DSS carcinogenic model mice at 13 weeks after the final DSS exposure.

## 2. Materials and Methods

### 2.1. Chemicals and Cell Culture

Fx-oil (5 w/v%) suspended in a palm oil containing slight dietary ingredients was provided by the Oryza Oil & Fat Chemical Co. Ltd. (Aichi, Japan). A Fx-free palm oil by the same company was used as a control-oil. All-*trans*-fucoxanthinol (FxOH) (purity, ≥98%) was kindly donated by Dr. Hayato Maeda (Hirosaki University, Japan). Azoxymethane (AOM) (purity, ≥95%) and dextran sodium sulfate (DSS) (M.W., 36,000–50,000) were purchased from Wako Pure Chemicals (Osaka, Japan) and MP Biomedicals (Solon, OH, USA), respectively. β-Actin, E-cadherin, pFAK(Tyr^397^) and integrin β1 antibodies were purchased from GeneTex (Irvine, CA, USA). pPaxillin(Tyr^31^) antibody was from Novex (San Diego, CA, USA). α-Smooth muscle actin (αSMA) and cleaved caspase-3 (Asp175)-Alexa488 antibodies were purchased from Cell Signaling Technology (Danvers, MA, USA). Goat anti-rabbit IgG-Alexa546 and ProLong Gold antifade reagent with DAPI were obtained from Invitrogen (Carlsbad, CA, USA). Human colorectal cancer DLD-1 cells and human fetal lung fibroblast MRC-5 cells were obtained from ATCC (Rockville, MD, USA) and RIKEN BRC (Tsukuba, Japan), respectively. These cells were grown in Dulbecco’s Modified Eagle’s Medium (DMEM) with 10% heat-inactivated fetal bovine serum (FBS), Glutamax (Invitrogen) and penicillin/streptomycin at 37 °C in a humidified atmosphere of 95% air and 5% CO_2_. All other chemicals and solvents were of high-grade quality.

### 2.2. Animal Experiments

An experimental protocol is shown in Figure 1B. ICR mice (male, five-week-old) were obtained from Sankyo Labo Service (Shizuoka, Japan). All mice were randomly assigned to four groups (5 mice/cage, 10 or 15 mice per group) and were maintained in the same humidity, temperature and 12 h light/dark cycle room. Solid food (Grade: MF, Oriental Yeast Co. Ltd.) and water were available ad libitum until sacrifice (about 16 weeks later). After a week of acclimation, mice were divided into group 1 (*n* = 15), 2 (*n* = 15), 3 (*n* = 10) and 4 (*n* = 10). Group 1 and 2 were given a single IP injection of AOM (10 mg/kg body weight (bw)) and received 3.0 w/v% DSS in drinking water for 1 week at 1 week after the AOM injection. Mice in groups 3 and 4 were injected with saline only (IP) and given normal water for 1 week. After the final DSS exposure, the mice in groups 1 and 2 were given solid food and water ad libitum until sacrifice (13 weeks) with control groups 3 and 4. Mice in groups 1 and 3 were given Fx oil (30 mg/kg bw) using a stomach sonde needle every 1 or 3 days during the final 3 weeks. Groups 2 and 4 were given the equivalent volume of control oil. The animals were inspected daily for clinical signs and mortality. Mice were sacrificed under isoflurane anesthesia. Subsequently, the large bowel of each mouse was washed with PBS, excised, cut open longitudinally and fixed in 10% phosphate-buffered formalin for at least 48 h. Tumor number and size on mouse colonic mucosa were analyzed under formalin permeation. The estimated tumor volume was expressed by using the formula of a (mm) × b^2^ (mm)/2 (a, long range: b, short range). Histopathological features were assessed on hematoxylin-eosin stained sections. The experimental and study design were approved by the Institutional Ethics Review Committee for Animal Experimentation in the Health Sciences University of Hokkaido (project identification code, 028a; date of approval, 7 March 2018), followed by ‘Guidelines for Animal Experiments in the Health Sciences University of Hokkaido’.

### 2.3. Fluorescence Immunohistochemistry on Anoikis-Inducing Cells

Paraffin-embedded tissue sections from the mice were stained with integrin β1 and cleaved caspase-3 (Asp175) by immunohistochemical techniques. Sections were de-waxed in xylene, washed in alcohol and distilled water. Antigen retrieval in the sections was performed by immersion in EDTA buffer (1 mM, pH 9) for 95 °C for 20 min. The sections were blocked at room temperature for 1 h with 5%BSA/0.1% polyoxyethylene (20) sorbitan monolaurate containing blocking solution (TBST) and then cooled to 4 °C. Subsequently, sections were probed with an integrin β1 primary antibody diluted 1:50 in 5%BSA/TBST overnight at 4 °C, then washed with TBST, and treated with goat anti-rabbit IgG-Alexa546 diluted 1:100 in TBST for 1 h at room temperature in the dark. Then, sections were washed with TBST, probed with cleaved caspase-3-Alexa488 primary antibody overnight at 4 °C. Sections were washed with PBS and mounted in ProLong Gold Antifade reagent with DAPI to detect cell nuclei. Co-localization of integrin β1 and cleaved caspase-3 were performed using a Nikon TE2000 microscope (Nikon Co. Ltd., Tokyo, Japan) equipped with EZ-C1 acquisition software. Cells having remarkably high fluorescence with cleaved caspase-3 protein (cleaved caspase-3^high^) and with low or negative expression of integrin β1 (integrin β1^low/-^) were analyzed comprehensively in adenocarcinoma tissues and in colonic mucosal crypt and evaluated as anoikis-like cells (integrin β1^low/-^/cleaved caspase-3^high^ cells), in contrast to non-anoikis-like integrin β1^high^/cleaved caspase-3^high^ cells, per set tissue area (mm^2^).

### 2.4. Western Blot

Tumors (≥2.0 mm at short diameter) were excised from mouse colonic mucosa in group 1 and 2 during dissection. Also, culture cells (see Materials and Methods Section 2.5) were harvested by trypsinization FxOH treatment. Tumors and cells were each washed twice with PBS and then total lysates were prepared by a lysis buffer. Fifty micrograms of cell proteins were subjected to western blot analysis using 10% acrylamide gels containing SDS. The protein-separated gels were then transferred to a PVDF membrane. The PVDF was blocked in 1% BSA/TBST at room temperature and was bound with each of the primary antibodies diluted 1:1000 in 1% BSA/TBST overnight at 4 °C. The membranes were washed and incubated with HRP-conjugated anti-mouse or anti-rabbit secondary antibodies diluted 1:5000 in TBST at room temperature for 1 h. The membranes were washed and subjected to chemiluminescence reagents.

### 2.5. Cell Viability

DLD-1 and MRC-5 cells were cultured at a density of 80 and 20 × 10^4^ cells on a 100-mm dish in 10%FBS/DMEM medium for 4 d, respectively. Then, DLD-1 cells were seeded at a density of 50 × 10^4^ cells into on a new 100-mm dish in the above conditions medium of the MRC-5 cells for 2 d, followed by repeated culture (total culture duration, 4d), i.e., DLD-1^MRC5^ cells. Control DLD-1 cells were cultured at the same cell concentration with fresh 10%FBS/DMEM medium for 4 d, i.e., DLD-1^None^ cells. Subsequently, both DLD-1^MRC5^ and DLD-1^None^ cells were seeded at a density of 5 × 10^4^ cells/ml into 24-well plates in 10%FBS/DMEM medium for 3.5 h. The adherent cells were incubated with 1%FBS/DMEM medium containing FxOH (final concentration, 1.0 or 5.0 μmol/l) or a vehicle (dimethylsulfoxide, DMSO) for 2 d. Then, cells were incubated with 25 μl of WST-1 reagent and measured using multiple ELISA reads (TECAN Japan, Tokyo, Japan).

### 2.6. Statistical Analysis

Values are the mean ± SE. Significant differences between the two groups were determined by Fisher’s exact probability test for incidence of colorectal lesions, Wilcoxon rank sum test, and one-way ANOVA with Tukey-Kramer post-hoc test for multiple comparisons, and the differences were considered statistically significant when *p* < 0.05 (*), *p* < 0.01 (**), *p* < 0.001 (***) and *p* < 0.00001 (*****).

## 3. Results

### 3.1. Histopathological Findings

Fx-oil was administered by gavage to mice for the final three weeks (Figure 1B). Unusual clinical signs were not seen at administration of Fx to mice in these periods. There was no significant difference in body weights among the four groups during the Fx treatment period (Figure 1C). The number of tumors in the Fx-treated group (group 1) significantly decreased compared to those of the control group (group 2); Fx-treated group 2.9 ± 1.3 and control 6.7 ± 1.5 (Figure 1D). No significant difference was observed in the tumor sizes between the Fx-treated group (23.3 ± 7.8 mm^3^) and the control group (19.6 ± 5.4 mm^3^) (Figure 1E). Pathological examination revealed that the incidence and multiplicity of mucosal damage (supposed to be a cured ulcer), dysplastic crypts, adenoma and adenocarcinoma significantly decreased and/or tended to be lower by Fx administration (Table 1, Figure 2A,B). Of note, the multiplicity of adenocarcinoma in the Fx-treated group (0.6 ± 0.3) was significantly decreased compared with that of the control group (2.4 ± 0.9) (Figure 1D).

### 3.2. Enhancement of Anoikis Induction in Colonic Adenocarcinoma and Mucosal Crypts by Fx

To evaluate the anoikis induction by Fx administration, anoikis-like integrin β1^low/-^/cleaved caspase-3^high^ cells were measured in colon adenocarcinoma and mucosal crypts in AOM/DSS mice by fluorescent immunohistochemistry. In addition, non-anoikis-like integrin β1^high^/cleaved caspase-3^high^ cells were also measured (Figure 3A and Figure 4A). As a result, anoikis-like cells in colorectal adenocarcinoma were significantly increased 8.3-fold by Fx administration in group 1 compared with those of the control. No significant difference was observed regarding non-anoikis-like cells in colorectal adenocarcinoma between groups 1 and 2. In addition, more anoikis-like cells were observed than non-anoikis-like cells in group 1. Meanwhile, fewer anoikis-like cells were observed than non-anoikis-like cells in group 2 (Figure 3B). No correlation was found between the number of cells per adenocarcinoma tissue area (mm^2^) and the whole area of adenocarcinoma (mm^2^) in either anoikis-like or non-anoikis-like cells examined in groups 1 and 2 (Figure 3C). Moreover, we investigated the location (upper and lower sites) of anoikis-like cells in colon adenocarcinoma. As a result, a difference in the number of cells between the upper and lower sites was not observed in either anoikis-like or non-anoikis-like cells in colon adenocarcinoma in group 1. However, the number of anoikis-like and non-anoikis-like cells on the upper site was both higher than the number of those cells on the lower site in group 2 (Figure 3D,E). On the colonic mucosal crypts, the number of anoikis-like cells in group 1 was significantly increased 3.5-fold by Fx administration compared with those of control mice in group 2, whereas no significant difference was observed among the four groups in non-anoikis-like cells (Figure 4B).

### 3.3. Protein Expression and Activation Related to Integrin Signaling in Colonic Normal Mucosa and Tumor by Fx

To evaluate the effect of Fx on protein expression and on activation levels involved in anoikis induction in AOM/DSS mice, the integrin signal-related protein plus αSMA, a CAF marker, was measured in normal colonic mucosa and tumors of the mice (Figure 5). Integrin β1 and αSMA expression, and pFAK (Tyr^397^) and pPaxillin (Tyr^31^) decreased 0.7-, 0.5-, 0.5- and 0.6-fold in normal mucosa in group 1 by Fx administration, respectively, but E-cadherin expression did not change, compared with those in group 2 (Figure 5B). 

Moreover, Integrin β1 (1.6-fold), E-cadherin (0.3-fold) and αSMA (26.6-fold) expression, and pFAK (Tyr^397^) (2.2-fold) and pPaxillin (Tyr^31^) (0.4-fold) altered in the tumors in group 2 compared with normal mucosa in group 1. Meanwhile, the expression and activation for these five proteins in tumors of group 1 were very similar to those of group 2 (Figure 5C).

### 3.4. Effect of Fucoxanthinol in DLD-1 Cells Using the Culture Supernatant of Fibroblast MRC-5 Cells

The protein profiles of DLD-1^None^ and DLD-1^MRC5^ cells were determined by western blot. In DLD-1^MRC5^ cells, the pFAK (Tyr^397^) (1.5-fold) and pPaxillin (Tyr^31^) (1.3-fold) were activated in comparison with those of DLD-1^None^ cells. Integrin β1 expression was almost the same between the two cell types (Figure 6A). The cell viability of untreated DLD-1^MRC5^ cells was 1.3-fold higher compared to the untreated DLD-1^None^ cells. Treatment with 1.0 and 5.0 μM FxOH inhibited the growth of both DLD-1^None^ and DLD-1^MRC5^ cells in a dose-dependent manner. Those growth suppressions were as follows: 1.0 μM FxOH for 38.1 ± 3.7% and 5.0 μM FxOH for 24.9 ± 2.3% against untreated DLD-1^None^ cells; 1.0 μM FxOH for 46.0 ± 4.9% (Ratio; 36.8%, vs. control DLD-1^MRC5^ cells) and 5.0 μM FxOH for 32.1 ± 3.0% (Ratio; 25.7%, vs. control DLD-1^MRC5^ cells) vs. untreated DLD-1^MRC5^ cells (125.1 ± 4.3%). Vehicle (DMSO) alone did not affect cell proliferation (Figure 6B).

## 4. Discussion

The present study demonstrated that Fx administration suppressed colonic lesions and increased anoikis-like integrin β1^low/-^/cleaved caspase-3^high^ cells, which were observed in both cancer cells in colonic adenocarcinoma and epithelial cells in colonic mucosal crypts of AOM/DSS mice. This is the first report suggesting anoikis induction by Fx treatment in colon adenocarcinoma, aimed at cancer chemoprevention.

We previously showed that Fx-oil administration every 1 or 3 days for 2 weeks increased anoikis-like integrin β1^low/-^/cleaved caspase-3^high^ cells in colonic mucosal crypts in AOM/DSS mice at 8 weeks after the final DSS exposure. This is the timing when few colon adenocarcinomas could be identified [31]. Therefore, in the previous experiment, we failed to obtain a statistical difference in the number of anoikis-like cells in adenocarcinoma between Fx-treated and control mice. However, the results led us to hypothesize that Fx-induced anoikis in normal mucosa might promote normal epithelial cell turnover to prevent development of neoplastic cells. In addition, Fx-induced anoikis in colon adenocarcinoma might also prevent prolongation of colon adenocarcinoma. In the present study, we investigated the effect of Fx-induced anoikis on colon adenocarcinoma development in AOM/DSS mice at 13 weeks after the final DSS exposure. Fx-oil was administrated every 1 or 3 days in the final 3 weeks of the experimental periods.

As a result, Fx administration significantly decreased the number of colonic tumors, the incidence and/or multiplicity of dysplastic crypts, adenoma and adenocarcinoma in the colon, compared with the those of control mice (Figure 1D, Figure 2A,B, and Table 1). As expected, anoikis-like integrin β1^low/-^/cleaved caspase-3^high^ cells were detected in colon adenocarcinoma by Fx treatment (Figure 3A). Interestingly, the number of anoikis-like cells in colonic adenocarcinoma in group 1 rose 8.3-fold compared with that of control mice in group 2, regardless of tumor size. No correlation was observed between groups 1 and 2 for the number of non-anoikis-like cells. These results suggest that anoikis induction by Fx would be a leading cause for cell death in adenocarcinoma in AOM/DSS mice. In addition, the number of anoikis-like cells in colon mucosa remarkably increased 2.0-fold in group 1, quite contrary to that of 0.3-fold in group 2 (Figure 3B,C). Moreover, the anoikis- and non-anoikis-like cells were observed ubiquitously in adenocarcinoma tissue in group 1. However, a number of anoikis- and nonanoikis-like cells were observed on the upper site in group 2 (Figure 3D,E). These facts suggest that anoikis induction in lower site of adenocarcinoma by Fx might be more enhanced than in upper site in group 1. On the other hand, on the colonic mucosal crypts, the number of anoikis-like cells in group 1 was significantly increased 3.5-fold by Fx administration, compared with that of control mice in group 2 (Figure 4A,B). The data in the colonic mucosal crypt were consistent with our previous data of anoikis enhancement observed in AOM/DSS mice by Fx administration [31]. Our results suggest that Fx administration induced anoikis more strongly in adenocarcinoma than in that of colonic mucosal crypts. Various molecular mechanisms of apoptosis induction by Fx have been demonstrated in culture cancer cells [24,25,26,27,28,29], however, no reports are on animal models. The present study would be the first report showing an anticancer mechanism by Fx in an animal cancer model.

So far, little evidence is available about anoikis in adenocarcinoma induced by natural and pharmaceutical compounds in humans and experimental animals. 4’-geranyloxy-ferulic acid and auraptene, both natural compounds isolated from plants, enhanced apoptosis in adenocarcinoma in AOM/DSS mice through suppression of NFκB, TNFα, NRF2, IL-6 and IL-1β [32]. In addition, collinin, a plant metabolite 8-methoxy type of auraptene, induces apoptosis in adenocarcinoma of AOM/DSS mice with attenuation of inflammatory factors, COX-2 and iNOS [33]. These three compounds might also increase anoikis-induced cells in adenocarcinoma because anti-inflammatory and anti-oxidative responses particularly accompanied the down-regulation of NFκB, TNFα and NRF2, the key pathways for anoikis [2,3]. Anoikis in colon adenocarcinoma might be frequently observed in carcinogenic animal models administrated with natural compounds, when carefully investigating the type of apoptosis.

Western blot analyses suggested remarkable anoikis induction in colonic mucosa of mice by Fx treatment, supported by data of the attenuation of integrin β1, pFAK (Tyr^397^) and pPaxillin (Tyr^31^), except for E-cadherin, in whole normal mucosa in group 1 compared with those of group 2 (Figure 5A,B). Although statistical difference on the protein change was unclear due to the small sample size (*n* = 2) on the tumor in group 1, the effect on whole tumor proteins by Fx seemed almost the same pattern as those in group 2 (Figure 5A,C). As integrin β1 and FAK proteins were highly expressed, not only in cancer epithelial cells but also in the major cells of a tumor microenvironment, including CAFs, myofibroblasts, immune cells and vascular cells [8,12], it may be difficult to assess the difference between groups 1 and 2. Thus, we evaluated the protein levels between normal mucosa in group 1 and tumors in group 2. As a result, integrin β1 expression and FAK activation were higher and paxillin activation and E-cadherin expression were lower in tumors in group 2 than in those of normal mucosa in group 1 (Figure 5A,C). As integrin β1 and FAK are positively, and E-cadherin is negatively associated with acquisition of the EMT phenotype in cancer cells, the cells in group 2 might show the EMT phenotype. [1,2,3]. Because paxillin is known to be positively associated with carcinogenesis with FAK and Src, the inactivation of paxillin in the tumors of group 2 remains unclear [34].

Notably, αSMA expression in normal mucosa and tumors in group 2 were 2.0- and 26.6-fold higher than that of normal mucosa of group 1, respectively (Figure 5B,C). The decrease of αSMA in normal mucosa by Fx is consistent with our previous study, in which low-dose Fx induced CAF reduction in colonic mucosa of AOM/DSS mice [35]. High expression of αSMA is a hallmark of CAF accumulation, and is considered a prognostic marker of human CRC [36,37]. CAFs are essential components of a tumor microenvironment and promote cancer stemness, tumor progression, alterations of ECM, tumor immunity and anoikis-resistance [1,38]. CAFs can sensitize to integrin subunits of CRC cells, and enhance cell migration through a CAF ligand, FGF2 [39]. In the present study, regardless of high αSMA expression in tumors of both groups 1 and 2 (Figure 5C), CAFs in group 1 increased 8.3-fold compared to those in group 2. Meanwhile, the number of anoikis-like cells in normal colonic epithelial cells of group 1 was 3.5-fold compared to those in group 2. 

We have identified an anti-proliferative effect of FxOH, a major biotransformation of Fx on human CRC cells, on DLD-1 cells strongly correlated with anoikis induction in a previous study [30]. Thus, in the next experiment, we investigated the anti-proliferative effect of FxOH on DLD-1 cells with activation of integrin signaling using a supernatant of medium cultured from human fibroblast MRC-5 cells. As a result, the anti-proliferative ratios obtained in DLD-1^None^ and DLD-1^MRC5^ cells were almost the same (Figure 6). This result suggests that FxOH is able to induce anoikis in cancer cells, regardless of activation of integrin signaling. Thus we assumed that anoikis enhancement in adenocarcinoma observed in AOM/DSS mice by Fx administration may be little influenced by CAF conditions.

The effect of Fx on cancer chemoprevention in AOM/DSS mice may not be explained by induction of anoikis. For instance, anchor-dependent cell aberration induces autophagy and entosis as well as anoikis [40,41]. Vitamin D and sulforaphane, functional lipids from food, enhance autophagy in small intestine tissue in *APC^1638N^* mice given a high-fat diet [42]. Nintedanib, a tyrosine kinase inhibitor, is suggested to induce entosis in human prostate cancer cells in xenografted mice, along with E-cadherin upregulation and CDC42 downregulation [43]. To examine the contribution of anoikis induction on cancer chemoprevention in AOM/DSS mice by Fx, further investigation by using a flow-cytometer is needed to determine the ratios of alive, anoikis, apoptosis with autophagy and entosis in adenocarcinoma tissue.

## 5. Conclusions

In summary, Fx administration inhibits tumorigenesis, and increased anoikis-like cells in colonic adenocarcinoma of an AOM/DSS mouse model. Interestingly, anoikis-like cells were observed ubiquitously in the colonic adenocarcinoma region of the Fx-treated mice, but was strongly observed in the upper region of the control mice. Fx also suppressed the expression and inhibited the activation of integrin signaling-related proteins in mucosal tissue in AOM/DSS mice. The in vitro experiments indicated that FxOH also suppressed cell growth in anchorage-enhanced DLD-1 cells. Our findings suggest that Fx can induce anoikis in cancer epithelial cells in a carcinogenic mouse model. 

## Figures and Tables

**Figure 1 jcm-09-00090-f001:**
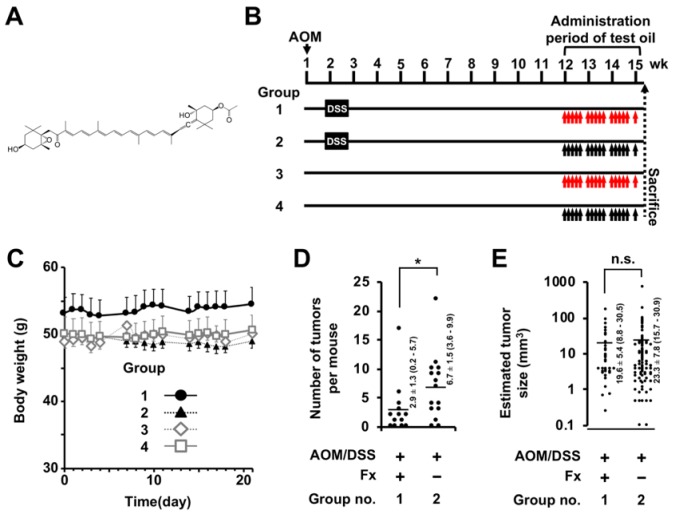
Body weight, tumor number and size in colitis-induced carcinogenesis model mice. (**A**) Chemical structure of fucoxanthin (Fx). (**B**) Experimental protocol for AOM/DSS induced colorectal carcinogenesis. Fx-oil was administrated to groups 1 and 3 at 30 mg Fx/kg bw every 1–3 days during the final 3 weeks before sacrifice (red arrows). The control groups with or without AOM/DSS treatment were given the equivalent volume (μl) of control-oil (Fx-free) (black arrows). (**C**) Body weight changes during the period of Fx administration. (**D**) Tumor number. Means ± SE (Fx, *n* = 13; control, *n* = 15). (**E**) Estimated tumor size. Means ± SE (Fx, *n* = 38; control, *n* = 101). The parentheses are 95% confidence intervals. Significant differences in tumor number (**D**) and estimated tumor size (**E**) were performed by Wilcoxon rank sum test (vs. group 2). * *p* < 0.05. AOM: Azoxymethane DSS: dextran sodium sulfate.

**Figure 2 jcm-09-00090-f002:**
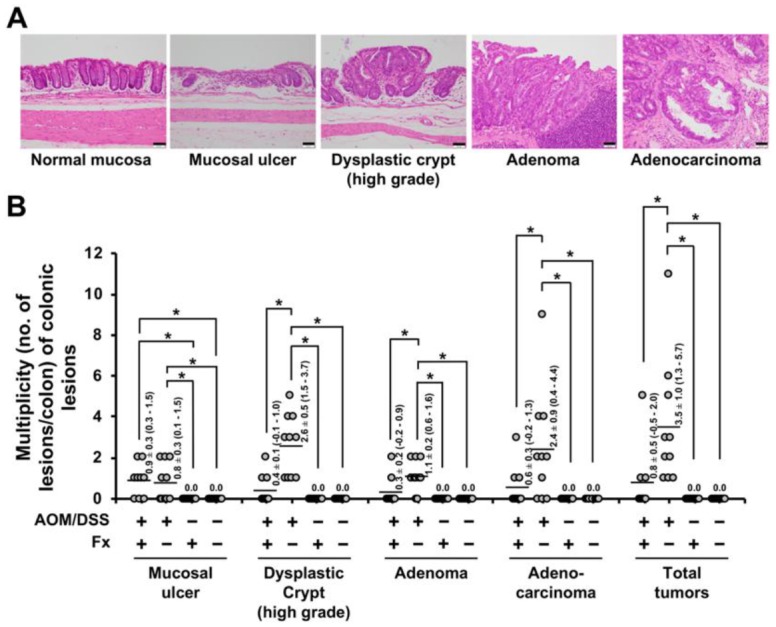
Multiplicity (no. of lesion/colon) of colonic lesions. (**A**) A representative photograph is shown for normal mucosa, mucosal damage, dysplastic crypt (high grade), adenoma and adenocarcinoma. Bar, 50 μm. (**B**) Number of colonic lesions per mouse. Averages of multiplicity (flat bars) are shown as a Mean ± SE (*n* = 9–10). The parentheses are 95% confidence intervals. Significant difference was performed by one-way ANOVA with a Tukey-Kramer post-hoc test. * *p* < 0.05. Individual colonic lesion was diagnosed by a pathologist (co-author, Takuji Tanaka).

**Figure 3 jcm-09-00090-f003:**
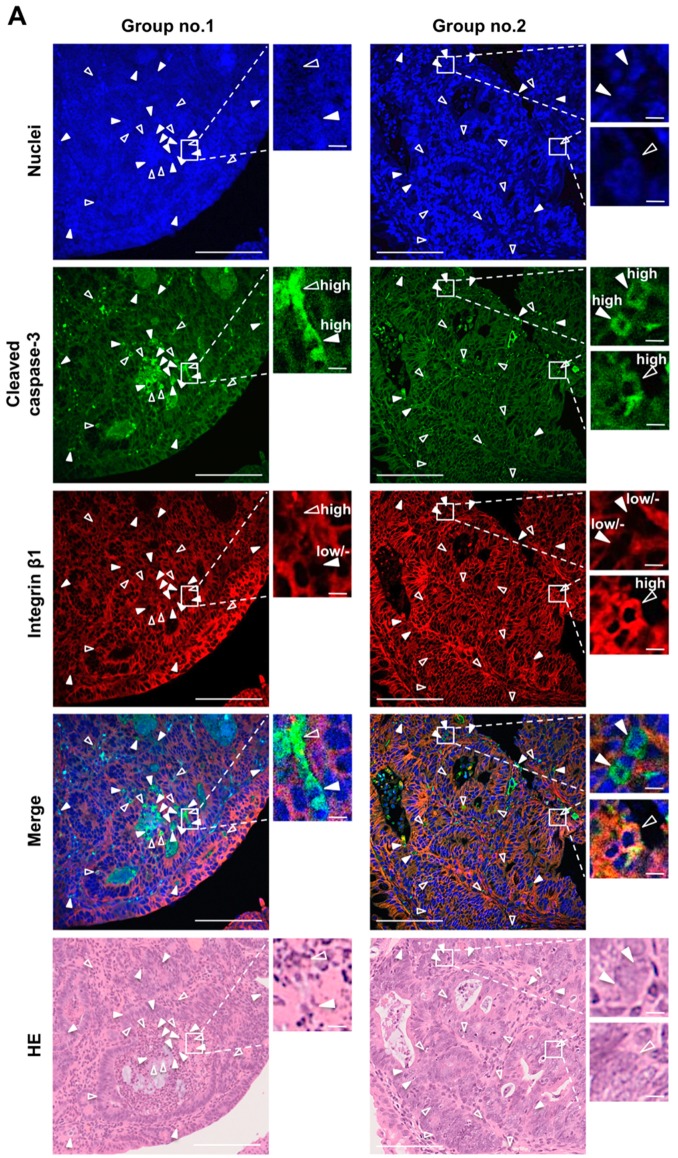
Detection of anoikis-like cells in colon adenocarcinoma of colitis-induced AOM/DSS mice with or without fucoxanthin (Fx) administration. The nuclei (blue fluorescence), cleaved caspase-3 (green fluorescence), and integrin β1 (red fluorescence) were observed by confocal microscopy. (**A**) Arrows show cells with negative/low expression (solid triangular arrows, anoikis-like cells) and high expression (open triangular arrows, non-anoikis-like cells) of integrin β1 plus high expression of cleaved caspase-3. HE, hematoxylin-eosin (HE)-stained sections. Long and short bars are 100 and 5 μm, respectively. (**B**) The number of anoikis-like cells (black box) and non-anoikis-like cells (white box) per tissue area (mm^2^) was determined by confocal microscopy. (**C**) The correlation plots between the cell number per tissue area and the whole area in adenocarcinoma. X-axis, the whole area of adenocarcinoma (mm^2^); Y-axis, the number of anoikis-like cells (black circle) and non-anoikis-like cells (white circle) per tissue area (mm^2^). (**D**) The rough compartment of upper and lower sites in adenocarcinoma. (**E**) The number of anoikis-like cells (black box) and non-anoikis-like cells (white box) per tissue area (mm^2^) in the upper or lower area in adenocarcinoma. Mean ± SE (group 1, *n* = 5; group 2, *n* = 20). Significant difference was performed by one-way ANOVA with a Tukey-Kramer post hoc test or Wilcoxon rank sum test. * *p* < 0.05 and ** *p* < 0.01.

**Figure 4 jcm-09-00090-f004:**
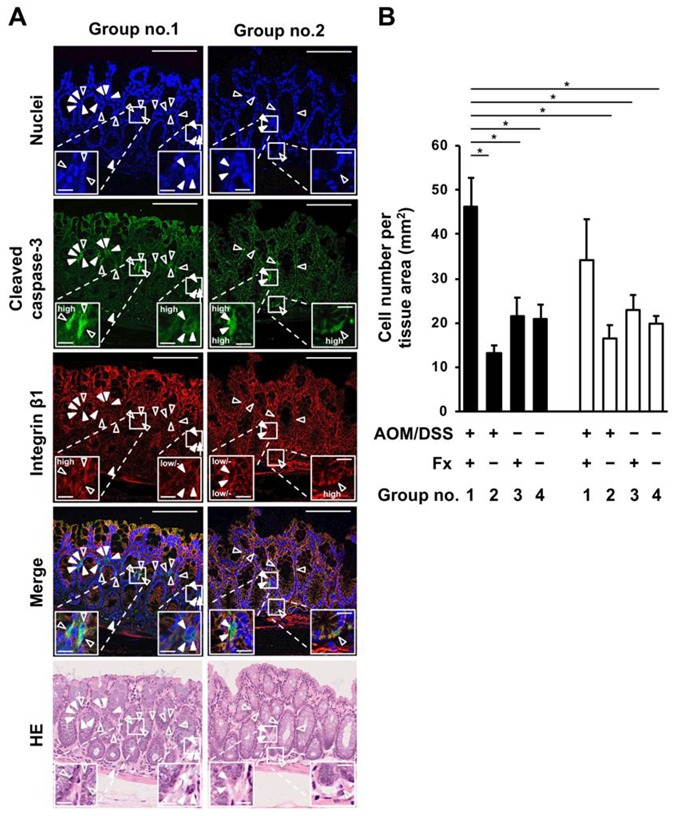
Detection of anoikis-like cells in colonic mucosal crypts in AOM/DSS mice with or without fucoxanthin (Fx) administration. The nuclei (blue fluorescence), cleaved caspase-3 (green fluorescence), and integrin β1 (red fluorescence) were observed by confocal microscopy. (**A**) Arrows show cells having negative/low expression (solid triangular arrows, anoikis-like cells) and high expression (open triangular arrows, non-anoikis-like cells) of integrin β1 plus high expression of cleaved caspase-3. HE, hematoxylin-eosin (HE)-stained sections. Long and short bars are 100 and 10 μm, respectively. (**B**) The number of anoikis-like cells (black box) and non-anoikis-like cells (white box) per tissue area (mm^2^) was determined by confocal microscopy. Mean ± SE (*n* = 5). Significant difference was performed by one-way ANOVA with a Tukey-Kramer post-hoc test. * *p* < 0.05.

**Figure 5 jcm-09-00090-f005:**
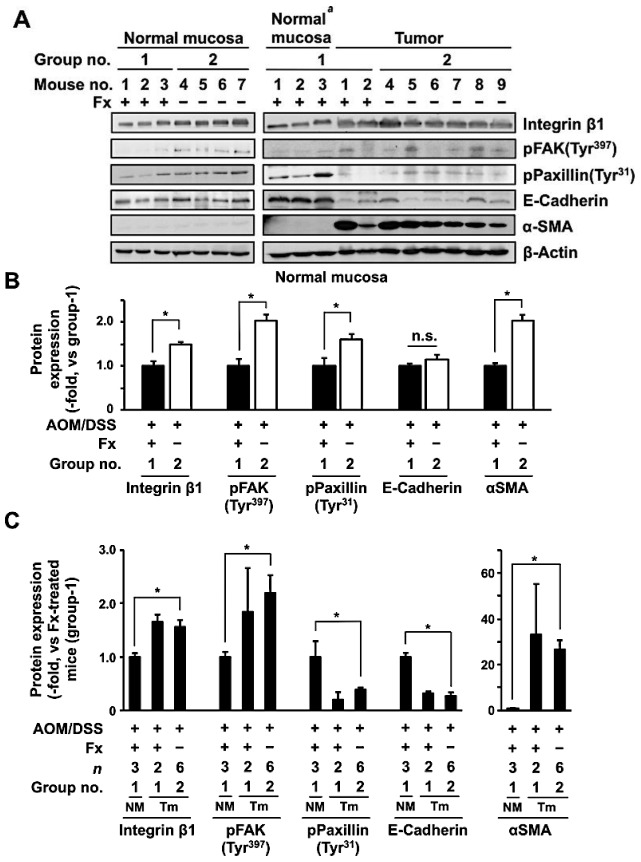
Protein expression in normal colorectal mucosa and tumor tissue from AOM/DSS mice with or without fucoxanthin (Fx) administration. (**A**) Normal colonic mucosa and tumors were collected, and indicated protein levels were evaluated by western blotting. The value of each band of (**B**) normal mucosa and (**C**) tumor was normalized to that of the β-actin band density from the image. (**B**) Each protein level in group 1 was set as 1.0. (**C**) Relative protein expression from tumor mucosa was quantified in comparison with normal mucosa of group 1. Mean ± SE (*n* = 3–6). Significant difference was performed by Wilcoxon rank sum test. * *p* < 0.05. ^a^ Sample no. 1–3 in normal mucosa in group 1 are the same samples as those in group 1 in the left panel. NM, normal mucosa. Tm, tumor mucosa.

**Figure 6 jcm-09-00090-f006:**
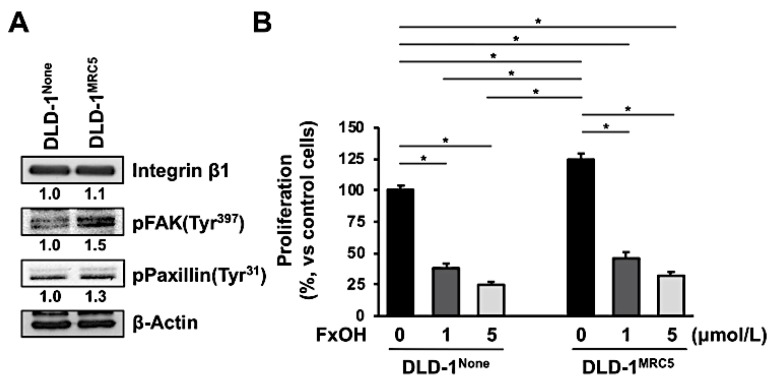
Antiproliferative effects of fucoxanthinol (FxOH) on DLD-1^None^ and DLD-1^MRC5^ cells. DLD-1^None^ and DLD-1^MRC5^ cells were treated with 1.0 or 5.0 μM FxOH for 2 d. (**A**) Untreated DLD-1^None^ and DLD-1^MRC5^ cells were collected, and their protein levels were evaluated by western blotting. The value of each band was normalized to that of the β-actin band density from the image. (**B**) Cell viability was determined using a WST-1 assay. Values are the mean ± SE (*n* = 3). Significant difference was performed by one-way ANOVA with a Tukey-Kramer post-hoc test. * *p* < 0.05.

**Table 1 jcm-09-00090-t001:** Incidence (%) of colonic lesions ^1^.

Group No.	Treatment	Mucosal Ulcer ^2^	Dysplastic Crypts	Adenoma	Adenocarcinoma	Total Tumors
1	AOM-DSS, Fx 30 mg/kg bw	67%	33% **	22% *	33%	33% **
2	AOM-DSS	50%	100%	80%	70%	100%
3	Fx 30 mg/kg bw	0% *	0% *****	0% ***	0% **	0% *****
4	None	0% *	0% *****	0% ***	0% **	0% *****

^1^*n* = 9–10. ^2^ It was detected as mostly healed. Significant difference from group 2 (* *p* < 0.05, ** *p* < 0.01, *** *p* < 0.001, ***** *p* < 0.00001) by Fisher’s exact probability test. Individual colonic lesion was diagnosed by a pathologist (co-author, Takuji Tanaka).

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
