# Peer review of "Dietary Fucoxanthin Induces Anoikis in Colorectal Adenocarcinoma by Suppressing Integrin Signaling in a Murine Colorectal Cancer Model"

_jcm, 2019, doi:10.3390/jcm9010090_

Round 1
Reviewer 1 Report
The report is well written and documents effects of Fx in a mouse model of colon carcinogenesis. Treatment with Fx shows signficant reduction of colon lesions as well as induction of anoikis in colonic epithelial cells. The investigation does not rule out other mechanisms responsible for the reduction in colon carcinogenesis. In particular, does Fx have anti-inflammatory properties and if so, have they been examined in this animal model of colon cancer. The conclusions should be carefully worded to improved the understanding.
Author Response
We express our deep thanks for your comments. With respect to write the discussion section in our first submission, we have investigated on the various anti-cancer and anti-inflammatory mechanisms of Fx in mouse adenocarcinoma as much as possible. However, no information is available on anti-cancer and anti-inflammatory mechanisms in adenocarcinoma and normal mucosa of animal models by Fx other than our findings [the present study and Ref. 31], while apoptosis inducing mechanisms by Fx and FxOH in culture cancer cells have been well elucidated such as inhibition of caspase activation, cell cycle promotion, mitochondrial function, DNA polymerase activity, PI3K/Akt, MAPK-, NFκB-, and Stat- signals [Ref 24-29]. Then, in accordance with the Reviewer’s comment, we have added a sentence to the discussion section (page 12, line 346-348 in the revised manuscripts).
In addition, the English language and grammar have been corrected by a native speaker once again.
Reviewer 2 Report
In this manual script titled "Dietary Fucoxanthin Induces Anoikis in Colorectal Adenocarcinoma through Suppressing Integrin Signaling in a Murine Colorectal Cancer Model", the authors want to demonstrate Fucoxanthin potentially induces Anoikis in colorectal adenocarcinoma in the AOM/DSS mouse model.
According to the previous studies published by the same research group (ref 30 and 31), the results have already demonstrated that fucoxanthin could induce anoikis in colorectal cancer cells and in the normal colon mucosa of AOM/DSS mouse model. The major issue in this manuscript is extended to the tumor lesion of the carcinogenic mouse model. Thus, it needs to be considered very carefully that the novelty of this manuscript is qualified to be published on JCM.
The following are several questions and opinions.
Historical findings The authors have to describe detail information in the method section. How do they check and confirm the colonic lesions? How do they calculate the incidence and multiplicity? For Fig. 1D, 1E, and 2B, dot plots could be more suitable (please refer to Ref 31). And it would be better to have a table contains the confidence intervals (95% CI) of tumor number, tumor size, incidence and multiplicity in each group. Anoikis cells and the related genes The IF images in Fig 3A and 4A are confusing. It is lack of HE staining as contrasts. It's hard to differentiate tumor lesions by morphology. It's also too crowded to find cells. Taking several high-power fields (with fewer cells) is good enough to demonstrate their concepts. To quantify protein level changes, flow cytometry could be a better choice than IF. The cleavaged Caspase-3 shown in these figures also suggested that just a minor proportion of cells underwent cell death, not only anoikis but also apoptosis. It would be essential to investigate the proportion of alive tumor cells, apoptosis, and anoikis. To evaluate FAK and Paxillin activation, the pFAK and pPaxillin need to be normalized to the total FAK and Paxillin, respectively.Author Response
1. Historical findings: The authors have to describe detail information in the method section. How do they check and confirm the colonic lesions? How do they calculate the incidence and multiplicity?
We express our deep thanks for your comments. Pathological examinations have been carried out by a pathologist (co-author, Takuji Tanaka) by using H&E stained tissue sections in the mice. The four types of colonic lesions were counted per mouse and the incidence and multiplicity (mean±SE) were calculated. As pointed out, we have modified the text in the legends shown in Table 1 and Figure 2 (Page 5, line 203-204; Page 6, line 212-213).
Fluorescence immunohistochemistry on adenocarcinoma and normal mucosa in the mice was carried out together with a pathologist, based on the information of colonic lesions in the HE stained section.
2. For Fig. 1D, 1E, and 2B, dot plots could be more suitable (please refer to Ref 31). And it would be better to have a table contains the confidence intervals (95% CI) of tumor number, tumor size, incidence and multiplicity in each group.
In accordance with the Reviewer’s comments, we have modified the Figures 1D and 1E and 2B (page 5, line 190; page 6, line 206 and 210). As pointed out, we have also added the confidence intervals to tumor number, tumor size and the multiplicity in colonic lesions directly to the Figures but not making a new table (page 5, lines 190 and 197-198; page 6, lines 206 and 211). There is little information about confidence intervals on the incidence (%) of colonic lesions in the carcinogenic model mice. Therefore, we would prefer not to change the phrasing on the incidence of the colonic lesions. Please let us know if you have any questions on it.
Again, we greatly appreciate the Reviewer’s comments guiding us to a more accurate paper.
3. Anoikis cells and the related genes. The IF images in Fig 3A and 4A are confusing. It is lack of HE staining as contrasts. It's hard to differentiate tumor lesions by morphology. It's also too crowded to find cells. Taking several high-power fields (with fewer cells) is good enough to demonstrate their concepts.
Thank you for your valuable comments. We have added the H&E stained images in Figures 3A and 4A (page 7, line 234; page 8, line 244; page 9, line 257 and 263). Because the cutting position in paraffin-embedded tissues is different between IF-utilized unstained section and HE section, the tissue histology in each IF and HE section was observed with a slight or rather shift in Figures 3A and 4A. In addition, peeling off were ubiquitously observed in the IF section, compared with histology of HE sections. On these differences, we would appreciate your attention.
In accordance with the Reviewer’s comments, more near-field of fluorescence images might be better for adenocarcinoma in the mice. However, as anoikis-like and nonanoikis-like cells in adenocarcinoma tissues were sparsely distributed in both group 1 and 2, it was hard to present the both cells with differences between group 1 and 2 into more near-field. Fold changed images from original images was necessarily added beside or into the original images in both Figures 3A and 4A.
Moreover, the imaging sizes in adenocarcinoma tissue in Figure 3A were useful to compare with the distribution of anoikis-like and nonanoikis-like cells in Figure 4A because the imaging sizes in adenocarcinoma in Fugure 3A were equal with that of colonic mucosal crypts in Figure 4A. Therefore, we believe that it is better to maintain the size of anoikis cells in cancer tissues.
However, because it’s important to follow Reviewer-2 thoughts, we have changed their images in Figure 3A and 4A to higher resolution, for easier viewing.
4. To quantify protein level changes, flow cytometry could be a better choice than IF. The cleavaged Caspase-3 shown in these figures also suggested that just a minor proportion of cells underwent cell death, not only anoikis but also apoptosis. It would be essential to investigate the proportion of alive tumor cells, apoptosis, and anoikis.
We investigated the number of anoikis and apoptosis cells in adenocarcinoma in the mice by using formalin fixed tissue. Therefore, in accordance with the reviewer’s comments, we could not investigate the rations of anoikis and apoptosis cells with alive cells in adenocarcinoma in the mice. However, as we calculated anoikis and apoptosis cell numbers per tissue at almost all areas of each adenocarcinoma tissue by using panel photography of confocal microscopy, we believe that the relative value in both anoikis and apoptosis cells between group 1 and 2 is correct as a phenomenon that is happening into adenocarcinoma tissue.
In accordance with the reviewer’s comments, next future, we will launch a mouse experiment again and would like to evaluate the ratios of alive, anoikis, apoptosis with others such as autophagy and entosis by flow cytometer. This explanation has been added to the Discussion (page 13, line 402-405 in the revised manuscript).
We express our deep thanks for your comments.
5. To evaluate FAK and Paxillin activation, the pFAK and pPaxillin need to be normalized to the total FAK and Paxillin, respectively.
Thank you for your valuable comments. Your comments are very important for clarification of molecular effects in anoikis induction in colorectal cancer cells by Fx and FxOH. However, unfortunately, we have used up the available cell samples for preparation of our manuscript.
In accordance with the reviewer’s comments, next future, we will launch a new in vitro experiment containing the evaluation of total FAK and Paxillin expression. In addition, we would like to elucidate molecular mechanisms targeting anchor proteins in colon cancer cells by Fx and FxOH by using the methods of overexpression and knockout.
We strongly appreciate again for the Reviewer’s comments, regarding on our paper.